# Emerging Respiratory Viruses of Cats

**DOI:** 10.3390/v14040663

**Published:** 2022-03-23

**Authors:** Andrea Palombieri, Federica Di Profio, Paola Fruci, Vittorio Sarchese, Vito Martella, Fulvio Marsilio, Barbara Di Martino

**Affiliations:** 1Laboratory of Infectious Diseases, Faculty of Veterinary Medicine, University of Teramo, 64100 Teramo, Italy; apalombieri@unite.it (A.P.); fdiprofio@unite.it (F.D.P.); pfruci@unite.it (P.F.); vsarchese@unite.it (V.S.); bdimartino@unite.it (B.D.M.); 2Laboratory of Infectious Diseases, Department of Veterinary Medicine, University of Bari Aldo Moro, 70010 Valenzano, Italy; vmartella@uniba.it

**Keywords:** emerging respiratory viruses, cats, etiology, epidemiology

## Abstract

In recent years, advances in diagnostics and deep sequencing technologies have led to the identification and characterization of novel viruses in cats as protoparviruses and chaphamaparvoviruses, unveiling the diversity of the feline virome in the respiratory tract. Observational, epidemiological and experimental data are necessary to demonstrate firmly if some viruses are able to cause disease, as this information may be confounded by virus- or host-related factors. Also, in recent years, researchers were able to monitor multiple examples of transmission to felids of viruses with high pathogenic potential, such as the influenza virus strains H5N1, H1N1, H7N2, H5N6 and H3N2, and in the late 2019, the human hypervirulent coronavirus SARS-CoV-2. These findings suggest that the study of viral infections always requires a multi-disciplinary approach inspired by the One Health vision. By reviewing the literature, we provide herewith an update on the emerging viruses identified in cats and their potential association with respiratory disease.

## 1. Introduction

Feline upper respiratory tract disease (URTD) is a common cause of morbidity in kittens, especially in overcrowded or stressful conditions. URTD results from a complex, multifactorial interaction of respiratory pathogens, stress, and animal susceptibility [1,2,3]. The clinical signs considerably vary in severity and include coughing, sneezing, nasal and ocular discharges, lethargy, difficult breathing, and in some cases, respiratory distress caused by bronchopneumonia and death. Although the case fatality rate is low, in many shelters the clinical signs of URTD are selection criteria for euthanasia, so the consequences for affected cats are profound. Multiple pathogens can contribute to URTD in kittens, and coinfections are common in overcrowded environments and contribute to increased disease severity [2]. Since the clinical signs caused by the different pathogens associated with this syndrome are similar, differential diagnosis is challenging. The complex multifactorial etiology of URTD involves viral and bacterial agents, acting either alone or synergically. Over the years, feline calicivirus (FCV) and feline herpesvirus-1 (FHV-1), often in conjunction with *Mycoplasma felis*, *Bordetella bronchiseptica* and *Chlamydia felis* (*C. felis*), have been identified as the main viral causes of URTD [2,4,5]. Vaccination against FHV-1 and FCV plays an important role in managing respiratory diseases [6], but despite the large use of the vaccines, URTD is still a major problem in peculiar scenarios such as multi-animal households and shelters [3].

In recent years, using advanced molecular techniques, screening of feline respiratory samples has identified novel viruses. Whether these orphan viruses have the ability to cause respiratory disease is still uncertain, although epidemiological studies and clinical investigations are gradually gathering precious information. Since coinfections are common, chiefly in multi-animal environments, observational studies are difficult to interpret, and experimental infections would be required to establish a causal link between a newly identified virus and respiratory disease, as well as an association between mixed infections and increased disease severity. Notably, cats can be also be infected with respiratory viruses affecting humans. Over the past two decades, there have been pandemics caused by severe acute respiratory syndrome coronavirus (SARS-CoV) [7] in 2002, H1N1 influenza virus in 2009 [8] and SARS-CoV type 2 (SARS-CoV-2) at the end of 2019 [9] and in all cases these viruses derived from animal reservoirs. The strict interaction between humans and companion animals has raised public health concerns on the potential risk of reverse zoonotic interspecies transmission. The aim of this review is to provide a general overview on novel viruses that have recently been identified in cats focusing in particular on their contribution to infection and coinfection in URTD. Furthermore, an update on SARS-CoV-2 and influenza viruses with regards to their evolving interaction with cats was reported.

## 2. Emerging Feline Parvoviruses: Bufaviruses and Chaphamaparvoviruses

The *Parvoviridae* family is a large and remarkably diverse group of viruses with linear single-stranded DNA genomes and nonenveloped icosahedral capsids. They infect a wide range of invertebrate and vertebrate animals, including humans. The family *Parvoviridae* was established in 1975 and divided into two subfamilies in 1993 to accommodate parvoviruses that infect vertebrate *(Parvovirinae)* and invertebrate (*Densovirinae*) hosts [10]. However, the recent discovery of divergent, vertebrate-infecting parvoviruses, has led to a significant taxonomic reorganization of the family with the introduction of the novel subfamily *Hamaparvovirinae* that encompasses divergent densoviruses and vertebrate-infecting parvoviruses [11,12]. Feline panleukopenia parvovirus (FPV) has long been the only known feline-pathogenic parvovirus. FPV may cause acute enteritis, severe dehydration and sepsis in cats due to lymphoid depletion and pancytopenia [13]. In recent years, the expanding use of broad-range consensus PCRs and sequence-independent metagenomic approaches in diagnostics and research has allowed for the identification and characterization of novel feline parvoviruses, genetically unrelated to FPV, initially designated as feline bufavirus (BuV) [14] and feline chaphamaparvovirus (FeChPV) [15]. The role of these newly discovered parvoviruses in the etiology of URTD has been addressed in a limited number of epidemiological studies [14,16].

BuVs were originally identified in 2012 in Burkina Faso in faecal samples from a child with acute gastroenteritis [17]. Since then, BuV-like viruses have been detected in several animal species, including dogs and cats [14,18]. Feline BuV (FBuV) was first identified in domestic cats in 2017 in Italy, in respiratory samples collected from animals with or without respiratory signs and in faecal specimens from cats with gastroenteritis [14]. On sequence analyses of the complete VP2-coding region, the newly feline parvoviruses showed the highest nt identity (99.5–99.9%) to canine BuV [18], currently classified in the novel species *Carnivore protoparvirus 3* (genus *Protoparvovirus*) [19]. Only one study has so far investigated the possible etiologic role of carnivore BuVs as respiratory pathogen of cats [14]. On molecular screening of 574 feline samples (respiratory and enteric), BuVs DNA was detected with an overall prevalence of 9.2% (53/574), suggesting that these novel protoparvoviruses are common component of the feline virome. In this investigation, analysis of 484 nasal and oropharyngeal swabs revealed the presence of BuV DNA with a rate of 10.5%. The virus was detected with higher frequency in animals with respiratory symptoms (7.3–25.5%) than in healthy (12.9–23.5%) [14]. When analyzing age distribution, the virus was more common in juvenile animals ≤1 year of age. Coinfections with FCV, FHV-1 and *C. felis* were also investigated, revealing a positive correlation in samples coinfected with BuV and *C. felis*. In the same study [14], BuV was detected with a prevalence five times lower (2.2%) in faecal specimens from cats with acute enteritis than in respiratory samples, suggesting that the virus was relatively infrequent in the enteric tract. However, in a recent investigation performed on diarrheic and healthy cats in China [20], BuV DNA was detected at high prevalence rate in the feces of cats suffering of acute enteritis (27.8%), whilst the detection rate in asymptomatic was 4.1%.

Overall, the impact of these novel protoparvoviruses on feline health and the target organs/district remain to be established. BuVs highly genetically related to the canine and feline strains were also detected in faecal specimens of clinically healthy foxes and wolves [21]. Carnivore BuV was initially identified from young dogs with respiratory signs [18], but subsequent studies revealed that this virus is also a common component of the canine enteric virome [18,22,23,24]. A positive association between BuVs infection and diarrhoea in dogs has been reported in a study performed in Shanghai (China). Viral DNA was detected in 42.15% of diarrheic faecal samples collected from animals with enteritis, but not in the asymptomatic control group [22]. Of interest, carnivore BuVs have been also found in sera from dogs with CIRD or acute enteritis [22,23] and BuV-like viruses have been identified in blood or spleen samples from non-human primates and shrews [25,26] and in the mesenteric lymph nodes from sea otters [27], suggesting the possibility of systemic infections. Also, information on the genetic heterogeneity of these viruses is still limited. Evidence is starting to suggest that carnivore BuVs are genetically heterogenous [24,28]. Sequence and phylogenetic analyses of the complete genome of three canine BuV strains have revealed that two strains, although possessing a well conserved NS1 gene, differed genetically in the VP2 (87.6–89.3% nt and 93.9–95.1% amino acid [aa] identities), with 24 distinctive aa residues mostly located in the variable regions (VRs) considered as important markers of host range and pathogenicity of parvoviruses [29,30]. On phylogenetic analysis, these two divergent carnivore BuV strains formed a distinct cluster/genotype [24]. This seems to mirror the genetic variability observed within human BuVs that are classified into at least three distinct VP2 genotypes [17,31,32], each one representing a distinct serotype [33]. Interestingly, in the VP2 capsid region the sequence diversity observed between the two canine BuV genotypes is approximately four times as much as the variation (6–7 aa changes) observed between FPV and CPV-2, and four to five times as much (5–6 aa changes) as the variation observed between the variants CPV-2a/b/c and the original CPV-2 strains. These few aa differences account for important antigenic and biological changes (e.g., host range shift in vivo and in vitro, affinity for receptors) among members of the *Carnivore protoparvovirus 1* species (i.e., FPV, CPV-2 and CPV-2 variants) [34]. Accordingly, further studies are necessary to elucidate if the VP2 aa changes may affect some biological properties of carnivore BuVs.

In addition to BuVs, novel parvoviruses genetically closest to members of the genus *Chaphamaparvovirus*, previously described under an unofficial umbrella term “Chapparvovirus” [11], have been recently identified in domestic carnivores [16,35]. Chaphamaparvoviruses (ChPVs) comprise a divergent group of parvoviruses whose ability is to infect vertebrate hosts despite being genetically more related to invertebrate-infecting parvoviruses [11,12]. The first identification of carnivore ChPVs was documented in Colorado (USA) in 2017 on deep sequencing of faecal samples collected from two dogs with haemorrhagic diarrhoea of unknown aetiology [35]. Following ICTV classification criteria (>85.0% aa identity in the NS1), the canine virus was classified within the novel species *Carnivore chaphamaparvovirus 1* (CaChPV-1) [12]. In subsequent molecular studies performed on canine faecal specimens [35,36,37], viruses genetically close to the American canine ChPV strains have been detected at low prevalence rates either in diarrheic (1.5–4.3%) or healthy dogs (0.0–1.6%) and no significant association with enteric disease was found. Similar results were also obtained when testing cats with (2/171) and without signs of acute gastroenteritis (0/378) [38]. In 2019 a novel ChPV was identified at high prevalence rate (47.0%) in feline faecal samples during an outbreak of acute gastro-enteritis in a multi-facility feline shelter in British Columbia (Canada) [15]. In the NS1 the feline ChPV (FeChPV) strains resulted genetically distant from CaChPV-1 of feline and canine origin (76.0–77.0% aa identities), representing a novel species designated *Carnivore chaphamaparvovirus 2.* High divergence to CaChPV-1 strains was also observed in the VRs of the VP capsid protein [37], with several aa changes located in the main sites involved in tissue tropism and receptor attachment of parvoviruses [39,40]. To date, epidemiological information on these newly discovered FeChPV is limited to only three studies [16,41,42]. In a case–control study performed in Italy [16], on screening of 89 feline faecal samples, FeChPV was detected with an overall prevalence of 16.9%. Also, a marked and significant difference in prevalence was observed between diarrheic (36.8%, 14/38) and healthy animals (2.0%; 1/51), confirming previous observations on the possible aetiologic role of FeChPVs as feline enteric pathogen [15]. Conversley, a low prevalence (3.7%) was observed when assessing respiratory (oropharyngeal and ocular) samples collected from juvenile household cats with detection rates respectively of 3.3% (6/183) in animals with URTD signs and 4.3% (6/140) in healthy animals, either alone or in coinfection with FCV, FHV-1 and *C. felis*. Similar detection rates were reported in a study conducted in Turkey [41]. By screening oropharyngeal samples collected from 70 healthy cats, FeChPVs DNA was found in 2.8% (2/70) of the animals tested. The potential role of FeChPVs in development of feline chronic kidney disease (CKD) has been recently investigated [42] by analyzing with PCR, with in situ hybridization and with immunohistochemistry, a total of 75 archival formalin-fixed paraffin-embedded kidney samples collected from immunocompromised and healthy cats with CKD. However, FeChPV DNA was not detected in the kidney samples.

Overall, based on the limited literature [15,16,41], carnivore BuVs and FeChPVs should be regarded as common components of feline virome. A possible association between URTD and carnivore BuVs has been hypothesized. Conversely, preliminary data seems suggest that FeChPV infection is more common in cats with enteritis. Detailed observational studies are required to address the possible implications of these viruses for feline health. Furthermore, generating a large dataset of complete genome sequences, involving cohorts of animals from different geographical areas, will be pivotal for a better understanding of the epidemiology and the genetic heterogeneity of these viruses.

## 3. Coronaviridae

The family *Coronaviridae* include a large and heterogeneous group of enveloped and roughly spherical viruses of 100–160 nm in diameter. Coronavirus (CoV) genome is a non-segmented, positive-sense, single-stranded RNA of 27–32 kb in length. At the capped 5′ end, two Open Reading Frames (ORFs) (ORF1a and ORF1b) encode 15–16 non-structural proteins. The ORFs located at polyadenylated 3′-end of the genome encodes four structural proteins, spike (S), membrane (M), envelope (E) and nucleocapsid (N), along with a set of accessory proteins, depending on the species [43]. Rapid evolution and high-frequency mutations are CoVs notable features, affecting viral antigenic profile, pathogenicity, host range, cell tropism and transmissibility. The genetic diversification of CoVs is driven by the accumulation of nucleotide substitutions due to the lack of an efficient proof-reading activity of the RNA dependent-RNA polymerase and by recombination events that take place in case of coinfections with other CoV strains (homologous recombination) or even with other RNA viruses (heterologous recombination) [44].

CoVs classified in the subfamily *Orthocoronavirinae,* genera *Alphacoronavirus* and *Betacoronavirus*, are responsible for infection in several mammalian species, mainly resulting in respiratory and enteric diseases [45,46]. Among alphacoronavirus, feline coronavirus (FCoV) (species *Alphacoronavirus-1*) causes infections in domestic and wild *Felidae*. Approximately 20–60% of domestic cats are seropositive, with rates reaching values of 90% in animal shelters or multi-cat households [47,48,49]. FCoV is primarily a pathogen of the gastrointestinal tract and replicates in the intestinal epithelial cells, with a faecal-oral transmission from cats that are either persistently or transiently infected. FCoV infections are often subclinical, but in some cases, they may cause acute and chronic diarrhoea, stunting of kittens or transient upper respiratory signs in newly infected kittens and cats, and faecal incontinence in persistently infected carrier cats [50,51]. Furthermore, about 7–14% of FCoV infected cats may develop feline infectious peritonitis (FIP), a systemic disease characterized by effusions in the body cavities (effusive or wet FIP) or pyogranulomatous lesions in organs (dry FIP), with a high mortality rate [49]. The key event in the pathogenesis of FIP is the switch in viral cell tropism from enterocytes to macrophages and monocytes [52], likely triggered by the accumulation of point mutations located in the S gene [53] or by deletion/insertion in the group-specific genes 3c [54,55], 7b [54] or 7a [56].

In the human host, several human coronaviruses (HCoV) have been identified, namely HCoV-229E, HCoV-OC43, HCoV-NL63 and HCoV-HKU1, all of which are associated with respiratory tract infections and with cold-like mild symptoms [57,58,59,60]. More recently, a novel CoV (Hu-PDCoV) has been detected in plasma samples of three Haitian children with acute undifferentiated febrile illness [61]. The complete genome sequence analysis demonstrated that the human PDCoV was highly genetically related (99.9%) to porcine deltacoronavirus strains detected in China and the USA. Furthermore, in the last two decades, novel hypervirulent betacoronaviruses, of zoonotic origin, have emerged. In 2002, the severe acute respiratory syndrome CoV (SARS-CoV) appeared in China [7] and nearly a decade after, the Middle East respiratory syndrome CoV (MERS-CoV) emerged in Saudi Arabia [62]. Both SARS-CoV and MERS-CoV induced severe pneumonia, with mortality rates of 10% and 30%, respectively [63], and anticipated the emergence in late 2019 of SARS-CoV-2, associated with COVID-19 respiratory disease [9]. Although the mortality rate of SARS-CoV-2 was only 2.4%, the high transmissibility of the virus enabled its quick spread globally, generating a pandemic [64]. The zoonotic transmission event likely occurred at a seafood and animal market in Wuhan, Hubei Province, China [65]. Bats and/or pangolins were suspected as the potential species of origin for the novel betacoronavirus based on the sequence homology with CoVs isolated from these animals [9,66,67].

The potential role of companion animals in the pandemic has been investigated intensively [68,69,70,71,72,73,74,75,76,77,78,79,80,81,82,83,84,85,86,87,88,89,90,91,92,93,94] because of their close contact with humans. The susceptibility of cats either under natural or experimental conditions to betacoronavirus infections was already demonstrated during the 2002–2003 SARS-CoV emergency [95,96]. SARS-CoV-2 is strictly related to SARS-CoV other than genetically, also at a biological level, sharing the same host receptor angiotensin-converting enzyme type 2 (ACE2), the main cellular receptor for viral attachment [97]. Feline ACE2 is highly effective in mediating SARS-CoV and SARS-CoV-2 infection based on in vitro virus-receptor binding studies [98,99]. Furthermore, the ACE2 gene in domestic cats is highly expressed in various tissues [100] included digestive (esophagus, rectum), respiratory (lung), and urinatory system (kidney), promoting the permissibility for infection.

After experimental inoculation in cats [90,91,92,93,94] a productive infection was observed, with viral shedding by both the oral and nasal route up to 5 days post-infection (dpi). Viral RNA was also detected in faecal swabs of the inoculated cats, with peaks of 2.29 log_10_ RNA copies/mL at 7 dpi [92,94]. In most studies, the cats did not develop clinical signs, although occasionally arching of the back and diarrhoea were observed [94]. Mild-to-moderate histopathological changes were found in nasal turbinates, trachea and lungs [90,92,93,94]. Characteristic changes in the lung tissues included thickened and multifocal alveolar septa, mild-to-moderate bronchiolitis with bronchiolar exudate, and accumulation of degenerative inflammatory cells, chiefly lymphocytes, monocytes, and neutrophils, around the blood vessels. Consistently with the histopathological findings, infectious viral particles and viral antigens were detected in nasal turbinates, trachea, soft palate, oesophagus, lungs and intestine [92,93,94]. In the experimental studies, cats developed a robust immune response against SARS-CoV-2, with neutralizing antibodies detected as early as 7 dpi and were resistant to re-infection upon subsequent challenge at 21 or 28 dpi [91,93]. Direct contact transmission to other cats was observed [93], although SARS-CoV-2 transmissibility and pathogenicity were significantly reduced by sequential passaging in cats [94].

In addition to domestic cats, naturally occurring SARS-CoV-2 infections have been described in several carnivore species, included wild *Felidae* [101], dogs [79], minks [102] and ferrets [103]. The first natural infection by SARS-CoV-2 related to human-to-cat transmission was documented in Belgium, in March 2020 [70]. A household cat of a COVID-19 patient tested positive for SARS-CoV-2 and showed self-limiting gastrointestinal and respiratory signs, characterized by troubled breathing and diarrhoea. Viral RNA was detected for about 10 days in oropharyngeal swab, vomitus and faeces. The presence of serum IgG during the convalescent-phase confirmed the active viral replication [70]. Another case of SARS-CoV-2 infection in cat was reported in Hong Kong. The pet was living with a COVID-19 patient and did not display any clinical signs, but the virus was detected in its respiratory secretions and feces [79]. On April 22, the OIE, the Centers for Disease Control and Prevention (CDC), and the United States Department of Agriculture (USDA) reported that two cats with respiratory signs (sneezing and nasal discharge) from New York State, USA, tested positive for SARS-CoV-2 by RT-qPCR. The cases were epidemiologically linked to suspected or confirmed human COVID-19 cases in their respective households [71]. To date, several SARS-CoV-2 natural infections of cats living in household with COVID-19 patients have been reported globally [104]. Clinical manifestations ranged from asymptomatic [72,73,74,75,76,77], to mild respiratory symptoms like sneezing, coughing, nasal and ocular discharge and conjunctivitis [71,78,80,81,82,83,84], to severe respiratory distresses associated with other unrelated diseases [85,86,87,88,89]. SARS-CoV-2–specific antibodies in cats have been reported on several occasions, with prevalence rates ranging from 4.5% to 43.8% [74,80,105,106,107,108,109] in animals from households in which family members have COVID-19. In contrast, during the first wave of the pandemic, the prevalence of anti-SARS-CoV-2 antibodies in cats without information regarding the potential exposure to SARS-CoV-2 was 0.69% in Germany [110], 0.76% in Croatia [111], 0.4% in the Netherlands [112] and 14.7% in Wuhan [113]. In a recent large survey [114] performed on convenience serum samples from cats (*n* = 956) collected from 48 states of the USA in 2020, the SARS-CoV-2 seropositivity was 0.4% (4/956). Seroprevalences are difficult to compare directly because of differences in the serological techniques. Cats in contact with COVID-19 patients have a 8.1-fold increase in the risk of being seropositive than cats in homes of unknown exposure [109]. Taken together, these findings support the hypothesis that cats may become infected if living in positive SARS-CoV-2 households. Although animal-to-human transmission of SARS-CoV-2 occurs in minks and mink-specific mutations have been reported [115], there is currently no evidence that cats can transmit infections to people, nor that cat-specific mutations or variants of SARS-CoV-2 may have developed [72,81]. The World Organization for Animal Health (OIE) and the CDC have released reports indicating that currently there is no evidence that pets may play a role in the spread of SARS-CoV-2 in the human population [116,117]. Based on current evidence, OIE does not recommend systematic testing of animals for SARS-CoV-2. Indeed, a strong and clear rationale and a risk assessment performed by Public Health and Veterinary Authorities should establish when sampling and testing animals would be necessary [116]. According to OIE definition, a case is confirmed in animals when SARS-CoV-2 is isolated from a sample or when viral nucleic acids are identified by targeting at least two genomic regions [118].

## 4. Influenza Viruses in Cats

Influenza viruses (family *Orthomyxoviridae*) cause highly contagious seasonal acute infection of the upper respiratory tract in humans. Symptoms associated with influenza virus infection vary from a mild respiratory disease confined to the upper respiratory tract to severe and in some cases lethal pneumonia, likely subsequent to secondary bacterial infections of the lower respiratory tract [119]. According to the World Health Organization, influenza annual epidemics cause 3–5 million cases of severe illness and 290,000 to 650,000 deaths [120]. Viruses belonging to the species *Influenza A Virus* (IAV) are responsible for both human seasonal epidemics and global pandemic outbreaks, with birds and pigs being recognized as primary reservoirs of infection [119]. Based on the major antigenic differences within the surface hemagglutinin (HA) and neuraminidase (NA) glycoproteins, IAVs are further classified into 18 HA and 11 NA subtypes. Thus far, more than 140 IAV subtype combinations have been identified in nature, primarily from wild birds.

Sporadic fatal disease due to natural IAV infection has been reported in various mammalian species, including domestic and wild carnivores [119]. So far, at least five subtypes of IAV have been reported in the literature as cause of acute respiratory illness in cats (H5N1, H1N1, H7N2, H5N6, H3N2). During the summer and early fall of 1996, a highly pathogenic avian influenza virus (HPAIV) subtype H5N1 (A/Goose/Guangdong/1/96) was detected in southern China [121]. The virus subsequently spread among poultry in Hong Kong. Despite strict control measures, the virus spread to many countries worldwide, resulting in high mortality in poultry and fatal infections in mammalian species, including humans [122,123,124]. The first natural infection by the H5N1 subtype in domestic cats was described in Bangkok (Thailand) in February 2004 in a fatal outbreak in 15 household cats with vomiting and coughing up blood. One of the cats had eaten a chicken carcass on a farm where there was an H5N1 virus outbreak. The presence of H5N1 virus was confirmed in three cats following necropsies. Intratracheal inoculation of a Vietnamese HPAIV H5N1 isolate and feeding of meat from infected birds to domestic cats confirmed their susceptibility, and the possibility of horizontal transmission in feline population [125]. Histologically, diffuse alveolar damage was observed in the animals, similar to the lesions observed in HPAIV H5N1-infected humans. Subsequently, single cases of H5N1 HPAI infections in cats from different parts of the world have been reported, mostly associated with recent avian outbreaks [126,127,128,129,130]. The first evidence on HPAIV H5N1 infection in domestic cats in Europe was reported in Germany in February 2006. Three cats were found dead in close spatiotemporal correlation with an outbreak of H5N1 in wild birds. In these areas, carcasses of wild swans, ducks, and geese had been accessible to both avian and mammalian scavengers. Therefore, infected wild birds were assumed to be the sources of infection for cats [130]. In the same year, three cats without apparent clinical signs tested positive for H5N1 in an animal shelter in Graz (Austria), after the introduction in the shelter of an infected swan [129]. The prevalence of antibodies for H5N1 IAV among different cat populations seems rather low (0.2–2.6%) [131,132], suggesting that IAV exposure in cats is rare and most often coincides with outbreaks in wild and domestic birds.

In early 2014, Sichuan province (Southern China), the first case of H5N6 IAV infection was reported in a man developing severe pneumonia after exposition to infected poultry [133]. In the same year, a fatal H5N6 IAV infection in a cat was documented in Northern China, where the virus spread due to extensive migration routes of wild birds [134]. In 2016, two additional strains were isolated from lungs from stray cats exhibiting high fever, loss of appetite, and lethargy in Zhejiang Province, Eastern China [135]. Sequence analysis suggested a reassortant origin of these strains receiving their genes from Chinese IAVs H5N6, H9N2 and H7N9. During 2016–2017 winter epidemics in domestic poultry and wild birds, H5N6 IAVs were isolated in three cats showing sudden clinical signs of salivation, lethargy, convulsion, and bloody discharge around the mouth and jaws. Cats died within 4 days after illness onset [136].

In 2005 in South Korea, three genetically similar H3N2 strains of avian origin were isolated from dogs showing severe respiratory signs [137,138,139]. The canine H3N2 was also detected in China in 2006 [140] and since then it has been repeatedly identified in those countries, indicating active IAV circulation in the Asian canine population. Other outbreaks have been reported in Thailand [141] and the United States, where the virus was introduced from South Korea via dogs rescued from live animal markets or meat production farms [142]. During March and April 2015, canine H3N2 virus was detected in dogs in shelters and kennels in the Chicago-area [142]. Mild to moderate respiratory signs were observed, often with a characteristic honking cough, with some progression to pneumonia but, generally, with few or no deaths [142]. Experimental challenge demonstrated that other animal species can be infected by the canine H3N2 virus, including ferrets, guinea pigs, and cats [143,144]. Severe respiratory disease was documented during a natural feline outbreak in South Korea in an animal shelter where both dogs and cats were co-housed. In cats, the infection was associated with tachypnoea, dyspnoea, lethargy, with high morbidity (100%) and mortality (40%) [145]. Dog-to-cat and cat-to-cat transmission of canine H3N2 IAV was also observed in a shelter in Gyunggido, South Korea [146]. Furthermore, several studies reported serological evidence of H3N2 IAV infections in cats [147,148,149,150]. In April 2009, a novel H1N1 IAV (pH1N1) was recognized as the cause of the flu pandemic in humans [8]. The isolate was identified as a novel swine-origin quadruple reassortant pH1N1, containing genes from Euro-Asiatic and American lineages of swine influenza, as well as avian and human influenza genes [151]. pH1N1 infection has been detected in companion animals since the fall of 2009 [152]. The first case of natural feline infection was reported in Iowa in November 2009 [153]. Since then, several feline infections have been documented worldwide. All the reported infections involved single-cat cases and were related to human-to-cat transmission. Susceptibility of cats to pH1N1 infection was also confirmed experimentally, demonstrating that the virus may cause respiratory disease in infected animals [154]. The clinical signs included high fever, depression, inappetence, severe dyspnoea, shallow and abdominal breathing, vomiting, conjunctivitis, oculo-nasal discharge, rhinorrhagia and in some cases death [153,155,156,157,158,159]. Specific antibodies against pH1N1 virus have been reported in cats, with prevalence rates ranging from 1.2% to 55.0% [148,157,159,160,161,162,163].

In November 2016, a severely ill cat showing clinical signs of respiratory disease was euthanized in a New York City animal shelter. Genome sequencing revealed that all the eight genes were genetically close to a lineage of H7N2 low-pathogenic avian IAVs that had been eradicated from poultry in 2006 [164]. Subsequent testing of animals in the same shelter identified widespread infection with H7N2 IAV among cats, while the virus was not found in dogs, chickens, or rabbits [165]. A human infection was observed in one of the veterinarians involved in the control program at the shelter, documenting the first case of cat-to-human transmission [165]. Outbreaks were reported in other shelters in New York and Pennsylvania [166]. A total of about 500 cats were found to be infected and only mild respiratory signs were documented [167].

Overall, domestic cats are considered naturally susceptible to many IAVs from other animal hosts. Cats may develop respiratory signs and histopathological lesions similar to those observed in humans. Although, the risk of cat-to-human transmission seems low [167], the epidemiology of these viruses should be thoroughly investigated either serologically or molecularly, since information on the IVs circulating in feline populations is still limited.

## 5. Discussion and Conclusions

Respiratory viruses remain a leading cause of disease in cats. In addition to well established viral agents (i.e., FCV and FHV-1) primarily involved in URTD, in recent years other viruses have been identified in the respiratory virome of cats, such as carnivore BuVs and ChPVs. However, virus discovery is only the first step and further investigations are surely required to clarify the potential clinical impact of these novel viruses on feline health and their possible role as respiratory pathogens. Also, a high prevalence of coinfections with viruses and bacteria has been observed in cats with URTD [2,4,5] and it would be necessary to understand whether mechanisms of synergisms are triggered during co-infections. The close social interactions of cats and humans in households provide a strong rational for studying the composition of the feline virome. Examples picturing the zoonotic and reverse zoonotic potential of IAVs and SARS-CoV-2 infections have been reported in several animal species, including domestic cats. This raise concerns on the possible implications for public health and, at the same time, requires a One Health envision in the study and management of infectious diseases of animals.

## Data Availability

Not applicable.

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
