# Peer review of "Emerging Respiratory Viruses of Cats"

_viruses, 2022, doi:10.3390/v14040663_

Round 1

Reviewer 1 Report

The presented review focuses on emerging respiratory viruses of cats. It is an interesting and in general well-written review.

In the first part on “Emerging feline parvoviruses: bufaviruses and chaphamaparvoviruses”, I would suggest focusing more in deep on the feline pathogenic aspect of these viruses on feline health. In my opinion, most of the first part of the review is spent describing potential respiratory viruses in general and how they affect other species than focusing on how these viruses really impact feline health or have pathogenic action for cats. The rest of the review is correctly most focused on feline health.

In general, I suggest to most expand knowledge from reviewed papers that reported data on the impact of the viruses on feline health.

Specific comments

Abstract – the abstract is too general in its content and should be more focused on the review content. I suggest adding in brief which novel feline respiratory viruses are covered in the review

Line 49. Please note that also SARS-COV-1 have some interaction with cats during the first outbreak in 2003, I suggest adding also brief information on this virus

Line 73-74: “The role of these newly discovered parvoviruses in the etiology of URTD 73 has been addressed in a limited number of epidemiological studies [13,15].” – Could the authors give more information on the studies in ref 13 and 15?

Line 86-89: “Analysis of 484 nasal and 86 oropharyngeal swabs, revealed the presence of BuV DNA with a rate of 10.5%. The virus 87 was detected with higher frequency in animals with respiratory symptoms (7.3% - 25.5%) than in healthy (12.9% - 23.5%), although this difference was not statistically supported [13]. When analyzing age distribution, the virus was more common in juvenile animals”. – I suggest to clearly state that this statement refers to cats

Line 114 (87.6-89.3% nt and 93.9%–95.1% aa identities) – please spell out the “aa” significance

Line 160-175: I like this part that treats in deep virus aspect on feline species

Line 233-234: could the authors explain why cats are susceptible to this human coronavirus, which are the mechanisms of host colonization in cats?

Author Response

Reviewer 1 (R1)

The presented review focuses on emerging respiratory viruses of cats. It is an interesting and in general well-written review.

R1.1 - In the first part on “Emerging feline parvoviruses: bufaviruses and chaphamaparvoviruses”, I would suggest focusing more in deep on the feline pathogenic aspect of these viruses on feline health. In my opinion, most of the first part of the review is spent describing potential respiratory viruses in general and how they affect other species than focusing on how these viruses really impact feline health or have pathogenic action for cats. The rest of the review is correctly most focused on feline health. In general, I suggest to most expand knowledge from reviewed papers that reported data on the impact of the viruses on feline health.

Reply to R1.1 – as suggested by Referee, in the revised manuscript, we modified some sentences posing more attention on the clinical impact that the novel bufaviruses have on feline health. However, we also included a sentence pointing-out that information on the epidemiology of this virus in cats with URTD is still limited to only one study [14]. More in detail, the paragraph referred to line 75-98 of the original manuscript was modified as follows “BuVs were originally identified in 2012 in Burkina Faso in faecal samples from a child with acute gastroenteritis [17]. Since then, BuV-like viruses have been detected in several animal species, including dogs and cats [14,18]. Feline BuV (FBuV) was first identified in domestic cats in 2017 in Italy, in respiratory samples collected from animals with or without respiratory signs and in faecal specimens from cats with gastroenteritis [14]. On sequence analyses of the complete VP2-coding region, the newly feline parvoviruses showed the highest nt identity (99.5 - 99.9%) to canine BuV [18], currently classified in the novel species Carnivore protoparvirus 3 (genus Protoparvovirus) [19]. Only one study has so far investigated the possible etiologic role of carnivore BuVs as respiratory pathogen of cats [14]. On molecular screening of 574 feline samples (respiratory and enteric), BuVs DNA was detected with an overall prevalence of 9.2% (53/574), suggesting that these novel protoparvoviruses are common component of the feline virome. In this investigation, analysis of 484 nasal and oropharyngeal swabs, revealed the presence of BuV DNA with a rate of 10.5%. The virus was detected with higher frequency in animals with respiratory symptoms (7.3 - 25.5%) than in healthy (12.9 - 23.5%) [14]. When analyzing age distribution, the virus was more common in juvenile animals ≤ 1 year of age. Coinfections with FCV, FHV-1 and C. felis were also investigated, revealing a positive correlation in samples coinfected with BuV and C. felis. In the same study [14], BuV was detected with a prevalence 5 times lower (2.2%) in fecal specimens from cats with acute enteritis than in respiratory samples, suggesting that the virus was relatively infrequent in the enteric tract. However, in a recent investigation performed on diarrheic and healthy cats in China [20], BuV DNA was detected at high prevalence rate in the faces of cats suffering of acute enteritis (27.8%), whilst the detection rate in asymptomatic was 4.1%.

Also, in the revised manuscript, the paragraph describing identification of chaphamaparvoviruses in other species (line 134-143 of the original manuscript) was deleted limiting information to the available literature on domestic carnivores.

 Specific comments

R1.2 - Abstract – the abstract is too general in its content and should be more focused on the review content. I suggest adding in brief which novel feline respiratory viruses are covered in the review.

Reply to R1.2 – as suggested by Referee, in the revised manuscript, Abstract was modified as follows “In recent years, advances in the diagnostics and deep sequencing technologies have led to the identification and characterization of novel viruses in cats as protoparviruses and chaphama-parvoviruses, unveiling the diversity of feline virome in the respiratory tract. Observational, epi-demiological and experimental data are necessary to demonstrate firmly if some viruses are able to cause disease, as this information may be confounded by virus- or host-related factors. Also, in recent years, researchers were able to monitor multiple examples of transmission to felids of viruses with high pathogenic potential, such as the influenza virus strains H5N1, H1N1, H7N2, H5N6 and H3N2, and, in the late 2019, the human hypervirulent coronavirus SARS-CoV-2. These findings suggest that the study of viral infections always requires a multi-disciplinary approach inspired to the One Health envision. By reviewing the literature, we provide herewith an update on the emerging vi-ruses identified in cats and their potential association with respiratory disease.

 R1.3 - Line 49. Please note that also SARS-COV-1 have some interaction with cats during the first outbreak in 2003, I suggest adding also brief information on this virus

Reply to R1.3 – many thanks to Referee for this observation. Information on SARS-CoV-1 was added in the introduction section (lines 48-51) of the revised manuscript “Over the past two decades, there have been pandemics caused by severe acute respiratory syndrome coronavirus (SARS-CoV) [7] in 2002, H1N1 influenza virus in 2009 [8] and SARS-CoV type 2 (SARS-CoV-2) at the end of 2019 [9] and in all cases these viruses derived from animal reservoirs.” A brief sentence on the susceptibility to SARS-CoV infection was also included in the paragraph Coronaviridae (line 285-287 of the revised manuscript) “The susceptibility of cats either under natural or experimental conditions to betacoro-navirus infections has been already demonstrated during the 2002–2003 SARS-CoV emergency [95,96].”

R1.4 - Line 73-74: “The role of these newly discovered parvoviruses in the etiology of URTD 73 has been addressed in a limited number of epidemiological studies [13,15].” – Could the authors give more information on the studies in ref 13 and 15?

Reply to R1.4 – all data reported in the manuscript on the possible association between the newly described parvoviruses and feline respiratory disease were referred to these only two epidemiological surveys [Diakoudi et al., 2019; Di Profio et al., 2021].

 R1.5 - Line 86-89: “Analysis of 484 nasal and 86 oropharyngeal swabs, revealed the presence of BuV DNA with a rate of 10.5%. The virus 87 was detected with higher frequency in animals with respiratory symptoms (7.3% - 25.5%) than in healthy (12.9% - 23.5%), although this difference was not statistically supported [13]. When analyzing age distribution, the virus was more common in juvenile animals”. – I suggest to clearly state that this statement refers to cats

Reply to R1.5 – In the revised manuscript, the sentence was rewritten explaining more clearly that molecular survey was performed on cat population (see also Reply to R.1.1).

 R1.6 - Line 114 (87.6-89.3% nt and 93.9%–95.1% aa identities) – please spell out the “aa” significance

Reply to R1.6 – In line 114 the “aa” significance was spell out.

 R1.7 - Line 160-175: I like this part that treats in deep virus aspect on feline species

Reply to R1.7 – Many thanks to the Referee for this observation.

 R1.8 - Line 233-234: could the authors explain why cats are susceptible to this human coronavirus, which are the mechanisms of host colonization in cats?

Reply to R1.8 – in the revised manuscript, the following sentence was added (line 287-294 of the revised manuscript) “SARS-CoV-2 is strictly related to SARS-CoV other than genetically, also at biological level, sharing the same host receptor angiotensin-converting enzyme type 2 (ACE2), the main cellular receptor for viral attachment [97]. Feline ACE2 is highly effective in mediating SARS-CoV and SARS-CoV-2 infection based on in vitro virus-receptor binding studies [98,99]. Furthermore, ACE2 gene in domestic cats is highly expressed in various tissues [100] included digestive (esophagus, rectum), respiratory (lung), and urinatory system (kidney), promoting the permissibility for infection.”

Reviewer 2 Report

In this review, the authors systematically described feline infectious viruses that cause upper respiratory tract diseases (URTD), including FCV and FHV-2 as the main viral causes of feline URTD, bufaviruses and chaphamaparvoviruses as emerging feline parvoviruses, SARS-CoV-2 and influenza viruses as zoonotic viruses, and FCoV which occasionally causees upper respiratory sign. Overall, this review is valuable to our studies in viral infections in companion animal cats.

Minor comments

  1. Line 87, “The virus was detected with higher frequency in animals with respiratory symptoms (7.3% - 25.5%) than in healthy (12.9% - 23.5%), although this difference was not statistically supported.” Without statistically significance or dramatic differences in ratios, such description should be avoided, especially for a review.
  2. Line 217, with respect to human coronaviruses, a recent identified zoonotic coronavirus, Hu-PDCoV, should be mentioned as well.
  3. In influenza viruses section, in addition to H5N1, H1N1, H7N2 and H3N2 subtypes, cat infections with H5N6 influenza viruses should be discussed.

Author Response

In this review, the authors systematically described feline infectious viruses that cause upper respiratory tract diseases (URTD), including FCV and FHV-2 as the main viral causes of feline URTD, bufaviruses and chaphamaparvoviruses as emerging feline parvoviruses, SARS-CoV-2 and influenza viruses as zoonotic viruses, and FCoV which occasionally causes upper respiratory sign. Overall, this review is valuable to our studies in viral infections in companion animal cats.

Minor comments

R2.1 - Line 87, “The virus was detected with higher frequency in animals with respiratory symptoms (7.3% - 25.5%) than in healthy (12.9% - 23.5%), although this difference was not statistically supported.” Without statistically significance or dramatic differences in ratios, such description should be avoided, especially for a review.

Reply to R2.1 – in the revised manuscript, the sentence “although this difference was not statistically supported” was removed.

R2.2 - Line 217, with respect to human coronaviruses, a recent identified zoonotic coronavirus, Hu-PDCoV, should be mentioned as well.

Reply to R2.2 – many thanks to the Referee for the observation. In the revised manuscript, the following sentence (line 265-269) was added “More recently, a novel CoV (Hu-PDCoV) has been detected in plasma samples of three Haitian children with acute undifferentiated febrile illness [61]. The complete genome sequence analysis demonstrated that the human PDCoV was highly genetically related (99.9%) to porcine deltacoronavirus strains detected in China and USA.”

R2.3 - In influenza viruses section, in addition to H5N1, H1N1, H7N2 and H3N2 subtypes, cat infections with H5N6 influenza viruses should be discussed.

Reply to R2.3 – in the revised manuscript, a paragraph on H5N6 was added (line 402-411) In early 2014, Sichuan province (Southern China), was reported the first case of H5N6 IAV infection in a man that developed severe pneumonia after exposition to in-fected poultry [133]. In the same year, a fatal H5N6 IAV infection in a cat was docu-mented in Northern China, where the virus spread due to extensive migration routes of wild birds [134]. In 2016, two additional strains were isolated from lungs from stray cats exhibiting high fever, loss of appetite, and lethargy in Zhejiang Province, Eastern China [135]. Sequence analysis suggested a reassortant origin of these strains receiving their genes from Chinese IAVs H5N6, H9N2 and H7N9. During 2016-2017 winter epidemics in domestic poultry and wild birds, H5N6 IAVs were isolated in three cats showed sudden clinical signs of salivation, lethargy, convulsion, and bloody discharge around the mouth and jaws, died within 4 days after illness onset [136].

Round 2

Reviewer 1 Report

I thank the authors for responding appropriately to  my comments and modifying the paper accordingly. In my opinion, the paper is now acceptable in present form for publication.

Author Response

R1.1. I thank the authors for responding appropriately to my comments and modifying the paper accordingly. In my opinion, the paper is now acceptable in present form for publication.

Reply to R.1.1. Thanks so much to the Referee 1 for his/her useful suggestions